# Short-Term Consumption of Cuban Policosanol Lowers Aortic and Peripheral Blood Pressure and Ameliorates Serum Lipid Parameters in Healthy Korean Participants: Randomized, Double-Blinded, and Placebo-Controlled Study

**DOI:** 10.3390/ijerph16050809

**Published:** 2019-03-05

**Authors:** Hye-Jeong Park, Dhananjay Yadav, Da-Jeong Jeong, Suk-Jeong Kim, Myung-Ae Bae, Jae-Ryong Kim, Kyung-Hyun Cho

**Affiliations:** 1Research Institute of Protein Sensor, Yeungnam University, Gyeongsan 712-749, Korea; hjgod6867@ynu.ac.kr (H.-J.P.); qpslquf@ynu.ac.kr (D.-J.J.); superiorgene@ynu.ac.kr (S.-J.K.); 2Department of Medical Biotechnology, Yeungnam University, Gyeongsan 712-749, Korea; dhanyadav16481@gmail.com; 3LipoLab, Yeungnam University, Gyeongsan 712-749, Korea; 4Drug Discovery Platform Technology Team, Korea Research Institute of Chemical Technology, Taejon 305-343, Korea; mbae@krict.re.kr; 5Department of Biochemistry and Molecular Biology, Smart-Aging Convergence Research Center, College of Medicine, Yeungnam University, Daegu 705-717, Korea; kimjr@med.yu.ac.kr

**Keywords:** policosanol, blood pressure, high-density lipoproteins, low-density lipoproteins

## Abstract

The current study was designed to investigate the short-term effects of policosanol consumption on blood pressure (BP) and the lipid parameters in healthy Korean participants with prehypertension. A total of 84 healthy participants were randomly allocated to three groups receiving placebo, 10 mg of policosanol, or 20 mg of policosanol for 12 weeks. Based on an average of three measurements of peripheral BP, the policosanol 20 mg group exhibited the most significant reduction, that is, up to 7.7% reduction of average systolic BP (SBP) from 136.3 ± 6.1 mmHg (week 0) to 125.9 ± 8.6 mmHg (week 12, *p* < 0.001). Between group comparisons using repeated measures ANOVA showed that the policosanol 20 mg group had a significant reduction of SBP at 12 weeks (*p* = 0.020) and a reduction of diastolic BP (DBP) at 8 weeks (*p* = 0.041) and 12 weeks (*p* = 0.035). The policosanol 10 mg and 20 mg groups showed significant reductions in aortic SBP of 7.4% and 8.3%, respectively. The policosanol groups showed significant reductions of total cholesterol (TC) of 9.6% and 8.6% and low-density lipoproteins (LDL-C) of 21% and 18% for 10 mg and 20 mg of policosanol, respectively. Between group comparisons using repeated measures ANOVA showed that the policosanol (10 mg and 20 mg) groups at 12 weeks had a significant reduction of TC (*p* = 0.0004 and *p* = 0.001) and LDL-C (*p* = 0.00005 and *p* = 0.0001) and elevation of %HDL-C (*p* = 0.048 and *p* = 0.014). In conclusion, 12-week consumption of policosanol resulted in significant reductions of peripheral SBP and DBP, aortic SBP and DBP, mean arterial pressure (MAP), and serum TC and LDL-C with elevation of % HDL-C.

## 1. Introduction

Hypertension is a key risk factor for the incidence of stroke and cardiovascular disease (CVD) and is often accompanied by dyslipidemia and diabetes [1,2]. It has been well established that the serum level of high-density lipoprotein-cholesterol (HDL-C) is inversely correlated with the incidence of CVD, diabetes, and Alzheimer’s disease [3,4,5]. However, the percentage of HDL-C in total cholesterol (TC), rather than amount of HDL-C (mg/dL), is considered a more important factor to predict the risk of incident hypertension [6]. Korean subjects with prehypertension have higher serum TC, triglycerides (TG), and serum TG/HDL-C [7]. In the same context, a study of Middle Eastern women showed that TG/HDL-C are strong predictors of incident hypertension [8]. A link between dyslipidemia, obesity and hypertension via stimulation of aldosterone synthesis has been proposed [9]. Recently, higher VLDL was shown to increase aldosterone production in mitochondria via binding of scavenger receptor-class B type I (SR-BI) and stimulation of several signaling pathways in acute aldosterone secretion and sustained aldosterone production [10]. Modified LDL, such as oxidized LDL and glycated LDL, also stimulate aldosterone release via Jak-2 activation for adrenocortical steroidogenesis [11].

Policosanol is mixture of eight aliphatic primary alcohols purified from sugar cane wax (*Saccharum officinarum* L.). These primary alcohols range from 24–34 carbon atoms, with octacosanol, triacontanol, dotriacontanol, hexacosanol and tetratriacotanol as main constituents. Consumption of policosanol can reportedly reduce TC and LDL-C with less oxidation of LDL [12,13]. In our previous studies, incorporation of policosanol into the core of HDL enhanced HDL functions via enhancement of anti-glycation, anti-apoptosis, and CETP inhibition [14]. Policosanol supplementation for 9 weeks in zebrafish had serum lipid-lowering and HDL-C-elevating effects via CETP inhibition; policosanol also ameliorated fatty liver changes [15]. Policosanol supplementation in Korean participants raised serum HDL-C and enhanced HDL functionality to inhibit oxidation and glycation of LDL and HDL [16]. Policosanol therapy for 8 weeks by healthy female subjects who had prehypertension resulted in lower blood pressure and CETP activity by elevating HDL/apoA-I contents and enhancing HDL functionalities, including cholesterol efflux and insulin secretion [17]. Eight weeks of policosanol supplementation in spontaneously hypertensive rats (SHR) resulted in remarkable decreases of blood pressure in a dose-dependent manner [18]. In addition to increasing the HDL-C level, long-term (24 weeks) consumption of policosanol lowered BP while enhancing the athero-protective functions of HDL as well as its antioxidant, anti-glycation, and anti-atherosclerotic activities [19].

Based on the previous studies, in the current study we aimed to analyze the effects of short-term (12 weeks) policosanol consumption on peripheral BP and central aortic BP along with any changes in the serum lipid profile in healthy Korean subjects who had prehypertension.

## 2. Materials and Methods

### 2.1. Policosanol

The policosanol raw material (also known as sugar cane wax alcohol) was obtained from Rainbow & Nature Pty, Ltd (Thornleigh, NSW, Australia). The policosanol consisted of several alcohol chains of various lengths. More than 90% of policosanol contents were higher aliphatic wax alcohols. Individual alcohols present in policosanol are identical with our previous report [19] and other reports for genuine policosanol [20]: namely, 1-tetracosanol (C_24_H_49_OH, 0.1–20 mg/g);1-hexacosanol (C_26_H_53_OH, 30.0–100.0 mg/g); 1-heptacosanol (C_27_H_55_OH, 1.0–30.0 mg/g); 1-octacosanol (C_28_H_57_OH, 600.0–700.0 mg/g); 1-nonacosanol (C_29_H_59_OH, 1.0–20.0 mg/g); 1-triacontanol (C_30_H_61_OH, 100.0–150.0); 1-dotriacontanol (C_32_H_65_OH, 50.0–100.0 mg/g); 1-tetratriacontanol (C_34_H_69_OH, 1.0–50.0 mg/g).

The placebo tablet had the same taste and smell with an identical colour. It contained the same basic pigment (Gardenia Blue color) and ingredients such as lactose, cellulose, glycerin fatty acid ester, magnesium stearate, etc, except for the policosanol.

### 2.2. Participants

Healthy male and female volunteers who had prehypertension (systolic 130–139 mmHg, diastolic 80–89 mmHg) were recruited by newspaper advertisement. All participants were pre-screened for suitability and the inclusion criteria were as follows: age 19–65 years old who had pre-hypertension without any history of endocrinological disorder. Heavy alcohol drinkers (>30 g ethanol (EtOH)/day) and those who had consumed prescribed drugs associated with hyperlipidemia, diabetes mellitus, or hypertension were excluded. All participants had unremarkable medical records without illicit drug use or past history of chronic diseases. On the first visit, all participants (*n* = 84) were grouped randomly by dice casting as group 1 (placebo), group 2 (policosanol 10 mg), or group 3 (policosanol 20 mg). All participants in each group consumed policosanol for 12 weeks, according to the study design (Figure 1). The Institutional Review Board at Yeungnam University (Gyeongsan, South Korea) approved the study and endorsed the protocol (IRB #7002016-A-2016-021), and the participants signed an informed consent form prior to the study beginning.

### 2.3. Study Design

This study was a double-blinded, randomized, and placebo-controlled trial with a 12-week treatment period. Participants were informed to consume one tablet per day containing policosanol (10 mg or 20 mg) or placebo, and film coated tablets were manufactured by CosmaxBio, Inc. (Jecheon, Korea) for this study. The ingredients, manufacturing process, and facility were approved by the Korean FDA. All participants were advised to avoid aggressive changes in dietary habits and excessive alcohol consumption (less than 30 g of ethanol per day). They were also instructed to avoid intense exercise (less than 30 min per day at 60–80% maximum capacity). If subjects had a sedentary lifestyle before the study, they were encouraged to balance their lifestyle during the policosanol consumption period to avoid bias due to exercise or other lifestyle habits.

### 2.4. Anthropometric Analysis 

On the visit day, anthropometrical parameters such as height, body weight, body mass index (BMI), subcutaneous fat (kg), and visceral fat mass (kg) were individually measured for each participant. The parameters were measured at the same time of day at 4-week intervals using an X-scan plus II body composition analyzer (Jawon Medical, Gyeongsan, Korea).

### 2.5. Measurement of Blood Pressure

Measurement of BP was carried out between 8 a.m.–12 p.m. on visit day because the participants had to do overnight fasting for blood collection. BP was measured using three measuring instruments at each visit and the average was recorded at 4-week intervals. First, we used a mercury sphygmomanometer for manual measurements by a licensed technician (S.-J.K.). Second, digital BP device (Omron HBP-9020, Kyoto, Japan) was employed. Third, SphygmoCor system (AtCor Medical, Sydney, Australia) was employed to measure peripheral and central aortic BP [21]. Central (aortic) BP provides more useful prognostic information than peripheral BP due to the proximity to important organs such as the heart, brain and kidneys. A meta-analysis of several longitudinal studies revealed that quantification of central aortic BP had good clinical significance and was a better predictor of cardiovascular events [22]. Using the SphygmoCor system, mean arterial pressure (MAP) was also measured, which defines the pressure during a single cardiac cycle, and was estimated using the formula (aortic SBP + aortic DBP x 2)/3.

### 2.6. Blood Analysis

After overnight fasting, blood was collected from each participant on the visit day. For plasma collection, blood was collected in a vacutainer (BD Biosciences, Franklin Lakes, NJ, USA) containing EDTA (final concentration of 1 mM) at weeks 0 and 12 by low-speed centrifugation (3000*g*) and stored at −80 °C until analysis. Total cholesterol (TC), triglyceride (TG), HDL-C, and glucose levels were measured in plasma using commercially available kits (Cleantech TS-S; Wako Pure Chemical, Osaka, Japan). Plasma aldosterone was measured by radioimmunoassay (RIA) using an instrument (1470-Gamma Counter, PerkinElmer (Waltham, MA, USA) at the Seegene Medical Foundation (Seoul, Korea). 

### 2.7. Data Analysis 

All values are expressed as the mean ± SD (Standard deviation) in the tables. SBP and DBP and blood profile were distributed normally as assessed by a Shapiro-Wilk test in Table 1 and Table 2. The differences in the placebo or policosanol 10 mg and policosanol 20 mg among the groups and over the follow up time were compared using repeated measures ANOVA with peripheral SBP, DBP and lipid profile, as listed in Table 3. When the ANOVA test for repeated measures was significant, the least significant difference (LSD) test was applied for post hoc pairwise multiple comparisons within the four paired means (0, 4, 8, and 12 weeks) and among the three groups (placebo, policosanol 10 mg, and policosanol 20 mg). The Student’s t-test for paired samples was used to compare the mean body composition, peripheral BP, aortic BP, and blood profile, as listed inTable 1, Table 2 and Table 4. The Student’s *t*-test (Table 1, Table 2 and Table 4) was used to compare the beginning of the study (at 0 week), which was used for homogeneity, with the middle of the study (4, 8, and and 12 weeks) to support the results of repeated measures ANOVA (Table 3 and Table 5).

In Figure 2, the data reveals a change in difference in all groups from a 100% initialized BP as mean ± (standard error of mean) SEM. A correlation study was performed using a Pearson’s test. Differences with a *p* value of <0.05 were considered significant and the statistical software SPSS (version 23.0; SPSS, Inc., Chicago, IL, USA) was used for statistical analysis. 

## 3. Results

### 3.1. Comparison of Body Composition and Peripheral BP

As shown in Table 1 and Table 2, there were no significant changes in BMI and body fat contents after 8- or 12-week consumption in all groups; values remained steady with BMI 22–24 kg/m^2^ and 1.6–2.1 kg of visceral fat. Based on the average of three methods of measurement of peripheral BP at week 8 (Table 1), the policosanol 20 mg group showed the greatest reduction of 6.1% and 6.0% for SBP and DBP, respectively, compared with week 0. SBP was reduced from 136.3 ± 6.1 mmHg (week 0) to 128.4 ± 9.1 mmHg (week 8, *p* < 0.001) and DBP was reduced from 84.2 ± 7.3 mmHg (week 0) to 79.5 ± 7.8 mmHg (week 8, *p* < 0.001). The policosanol 10 mg group showed 4.5 % and 3.6% reduction of SBP and DBP, respectively, at week 8 compared with week 0. The placebo group showed no change in brachial SBP and DBP after 8 weeks. We also quantified the confidence interval (CI, usually 95%) of blood pressure values in the different groups to measure the range that provides the variability of the observed population statistic (precision) and to report the probable relationship in the population from which the data were extracted (accuracy). For the placebo group, the measured CI at week 0 and week 8 for the peripheral and aortic pressure were SBP 95% (−0.24, 6.42), (1.68, 10.2) and DBP 95% (−1.78, 3.87), (−0.08, 5.12). For the policosanol 10 mg group, the measured CI at week 0 and week 8 for the peripheral and aortic pressure were SBP 95% (1.81, 10.2), (4.26, 12.4) and DBP 95% CI (0.12, 5.96), (0.40, 6.76). For the policosanol 20 mg group, the measured CI at week 0 and week 8 for the peripheral and aortic pressure were SBP 95% (5.53, 10.26), (3.54, 9.76) and DBP 95% (2.85, 6.52), (1.28, 7.06).

As shown in Table 2, the policosanol 20 mg group exhibited the most remarkable decrease in average SBP from 136.3 ± 6.1 mmHg at week 0 to 125.8 ± 8.7 mmHg at week 12 (7.7% reduction; *p* < 0.001). The policosanol 10 mg group showed a 6.1% reduction in SBP from 135.8 ± 12.3 mmHg at week 0 to 127.7 ± 9.6 mmHg at week 12 (*p* < 0.001), while the placebo group showed no change in BP between week 0 (134.4 ± 8.8 mmHg) and week 12 (132.1 ± 10.2 mmHg). For the placebo group, the measured CI at week 0 and week 12 for the peripheral and aortic pressure were SBP 95% (−1.52, 6.13), (−0.077, 8.83) and DBP 95% (−0.65, 5.28), (−0.38, 6.86). For the policosanol 10 mg group, the measured CI at week 0 and week 12 for the peripheral and aortic pressure were SBP 95% (4.25, 11.91), (5.46, 12.61) and DBP 95% (0.44, 6.96), (−0.817, 6.81). For the policosanol 20 mg group, the measured CI at week 0 and week 12 for the peripheral and aortic pressure were SBP 95% (8.06, 12.96), (7.31, 12.6) and DBP 95% (3.72, 8.55), (1.20, 7.83).

As shown in Table 3, from repeated measures ANOVA with SBP data, the policosanol groups (10 mg, 20 mg) showed significant differences from the placebo in point of time (*p* < 0.001) and the time and group interaction (*p* = 0.012). Further analysis with repeated measures ANOVA showed that both policosanol 10mg (*p* < 0.001) and 20 mg (*p* < 0.001) groups showed significant differences within the group, whereas the placebo group showed no significance within the group (*p* = 0.286). One-way ANOVA between the groups showed no significant difference among the three groups at 0 week (*p* = 0.753), 4 weeks (*p* = 0.770), and 8 weeks (*p* = 0.631), except for 12 weeks (*p* = 0.020). Using the least significant difference (LSD) method as a post-hoc test with the 12-week data, the policosanol 20 mg group showed a significant difference compared to the placebo group, whereas the 10 mg group showed no difference. 

Within each policosanol group, multiple comparisons using the LSD method revealed the policosanol 10 mg group to show a significant reduction of SBP at 4 weeks (*p* = 0.0004), 8 weeks (*p* = 0.007), and 12 weeks (*p* = 0.0002) compared to that at 0 week. The policosanol 10 mg group also showed a significant decrease in SBP at 12 weeks (*p* = 0.025) compared to that at 4 weeks. The policosanol 20 mg group showed a more significant reduction of BP at 4 weeks (*p* < 0.001), 8 weeks (*p* < 0.001), and 12 weeks (*p* < 0.001) compared to that at 0 week. The policosanol 20 mg group also showed a significant reduction of BP at 8 weeks (*p* = 0.021) and 12 weeks (*p* < 0.001) compared to that at 4 weeks. In addition, the policosanol 20 mg group showed a significant reduction of SBP at 12 weeks (*p* = 0.007) compared to that at 8 weeks.

From repeated measures ANOVA with DBP data (Table 3), the policosanol groups (10 mg and 20 mg) showed a significant difference from the placebo in the point of time (*p* < 0.001) and time and group interaction (*p* = 0.023). Further repeated measures ANOVA revealed significant differences between the policosanol 10mg (*p* = 0.028) and 20mg (*p* < 0.001) groups, while placebo group showed no significance within group (*p* = 0.270). One-way ANOVA analysis between the group showed that the three groups were similar at 0 week (*p* = 0.831) and 4 weeks (*p* = 0.532). On the other hand, the same analysis revealed a significant difference at 8 weeks (*p* = 0.041) and 12 weeks (*p* = 0.035). Using LSD method as a post-hoc test with the 8-week and 12-week data, the policosanol 20 mg group showed significant differences compared to the placebo group. 

Within the each policosanol group, multiple comparisons using the LSD method showed that the policosanol 10 mg group had a significant reduction of DBP at 4 weeks (*p* = 0.021), 8 weeks (*p* = 0.042), and 12 weeks (*p* = 0.028) compared to that at 0 week. The policosanol 20 mg group showed a more significant reduction of DBP at 4 weeks (*p* = 0.002), 8 weeks (*p* < 0.001), and 12 weeks (*p* < 0.001) compared to that at 0 week. The policosanol 20 mg group also showed a significant reduction of DBP at 8 weeks (*p* = 0.002) and 12 weeks (*p* < 0.001) compared to that at 4 weeks. These results suggest that 20 mg of policosanol is significantly more effective in lowering the SBP and DBP than 10 mg of policosanol with dose responsiveness according to repeated measures ANOVA. 

Figure 2 depicts the changes in the peripheral blood pressures (SBP and DBP 100 % initialized) during 12 weeks consumption of policosanol 10 mg, 20 mg, and placebo group. Between group comparisons using the LSD method, as a post-hoc test, showed that the policosanol 20 mg group had a significant reduction of SBP at 12 weeks up to −7.7% (*p* = 0.020) and a reduction of DBP at 8 weeks (−5.5%, *p* = 0.041) and 12 weeks (−7.1%, *p* = 0.035) compared with placebo. In contrast, the policosanol 10 mg group showed no significant difference compared with placebo at each time point, although the 10 mg group showed significant reduction of SBP and DBP from baseline of 0 week.

### 3.2. Change in Aortic BP and Mean Arterial Pressure

Based on the SphygmoCor measurements, the placebo group did not show significant changes in aortic SBP and DBP during the 12-week consumption. However, between 0 and 8 weeks, the placebo, policosanol 10 mg, and 20 mg groups showed aortic SBP reductions of 4.8%, 6.9%, and 5.6%, respectively. Aortic DBP was reduced significantly in policosanol 10 and 20 mg groups by 4.0% and 4.7% respectively, at week 8. Mean arterial pressure (MAP) was reduced in the policosanol groups in a dose-dependent fashion. In the policosanol 10 mg and 20 mg groups, MAP was reduced by 4.1% and 6.0%, respectively, while the placebo group showed no change (Table 1).

In week 12, as shown in Table 2, the placebo group had a similar aortic SBP: 122.1 ± 11.8 mmHg (week 0) and 117.8 ± 9.0 mmHg (week 12). In the policosanol 20 mg group, aortic SBP was reduced by 8.3% from 120.7 ± 8.7 mmHg (week 0) to 110.7 ± 9.0 mmHg (week 12, *p* < 0.001). The policosanol 10 mg group showed also significant reductions by 7.4% in aortic SBP from 123.0 ± 12.5 mmHg (week 0) to 113.9 ± 8.6 mmHg (week 12, *p* < 0.001). Although aortic DBP was significantly reduced only in the policosanol 20 mg group by 5.1% reduction, however, MAP was significantly reduced by 5.2% and 7.5% in the policosanol 10 mg and 20 mg groups, respectively. Interestingly, the peripheral SBP and DBP, aortic SBP and DBP, and MAP were lowered in a policosanol dose-dependent manner. 

### 3.3. Change in Lipid and Aldosterone Profiles 

After 12 weeks, as shown in Table 4, TC in the policosanol groups was significantly reduced by 9.6% and 8.6% by 10 mg and 20 mg, respectively, compared with the TC level in week 0 (Table 4). The placebo group showed an 8.3% increase in TC at week 12. Serum TG and glucose level were not changed in any groups after 12 weeks. However, serum HDL-C was significantly increased by 16% and 20% in the 10 and 20 mg policosanol groups, respectively, while the placebo group was not increased significantly. 

The percentage of HDL-C in TC (%HDL-C/TC) was also remarkably elevated in the policosanol groups in a dose-dependent manner. In the policosanol 10 mg group, %HDL-C/TC was increased to 25.9 ± 6.5% in week 12 compared with 20.2 ± 4.7% in week 0. In the policosanol 20 mg group, %HDL-C/TC was increased to 26.6 ± 6.0% at week 12 from 21.3 ± 8.4% at week 0 (Table 4). TG/HDL-C level was significantly reduced in the policosanol groups in a dose-dependent manner, while the placebo group displayed no change. In the policosanol 10 mg group, TG/HDL-C decreased to 3.0 ± 2.5 in week 12 from 3.2 ± 2.7 in week 0, while policosanol 20 mg decreased TG/HDL-C to 2.4 ± 1.6 in week 12 from 3.4 ± 3.6 in week 0. LDL-C level increased by 10% in the placebo group after 12 weeks, while the policosanol groups showed remarkable decreases of 20% and 18% in LDL-C for the 10 and 20 mg groups, respectively. 

As shown in top panel of Table 5, from repeated measures ANOVA with TC data, the policosanol groups (10 mg, 20 mg) showed only significant differences from the placebo in point of time × group interaction (*p* = 0.009). Although there were no difference in group and time, however, the time × group interaction was significantly different. Further analysis with repeated measures ANOVA showed that both policosanol 10 mg (*p* = 0.010) and 20 mg (*p* = 0.024) groups showed significant differences within the group, whereas the placebo group showed no significance within the group (*p* = 0.203). One-way ANOVA between the groups showed no significant difference among the three groups at 0 week (*p* = 0.915), 4 weeks (*p* = 0.169), and 8 weeks (*p* = 0.167), except for 12 weeks (*p* = 0.0005). Using the least significant difference (LSD) method as a post-hoc test with the 12-week data, the policosanol 10 mg group (*p* = 0.004) and 20 mg group (*p* = 0.001) showed a significant difference compared to the placebo group. Within each policosanol group, multiple comparisons using the LSD method revealed the policosanol 10 mg group to show a significant reduction of TC only at 12 weeks (*p* = 0.001) compared to that at week 0. The policosanol 10 mg group also showed a significant decrease in TC at 12 weeks (*p* = 0.010) compared to that at 4 weeks. The policosanol 20 mg group showed a more significant reduction of TC at 4 weeks (*p* = 0.005), 8 weeks (*p* = 0.015), and 12 weeks (*p* = 0.029) compared to that at week 0. These results suggest that policosanol groups showed significant reduction of TC with time dependent manner, while placebo group was not. 

As shown in middle panel of Table 5, from repeated measures ANOVA with LDL-C data, the policosanol groups (10 mg, 20 mg) showed significant differences from the placebo in point of group (*p* = 0.035), time (*p* < 0.001), and time × group interaction (*p* < 0.001), suggesting that LDL-C was significantly and distinctly reduced in policosanol groups. Further analysis with repeated measures ANOVA showed that both policosanol 10mg (*p* = 0.001) and 20 mg (*p* < 0.001) groups showed significant differences within the group, whereas the placebo group showed no significance within the group (*p* = 0.103). One-way ANOVA between the groups showed no significant difference among the three groups at 0 week (*p*=.916), 4 weeks (*p* = 0.226), and 8 weeks (*p* = 0.148), except for 12 weeks (*p* < 0.001). Using the least significant difference (LSD) method as a post-hoc test with the 12-week data, the policosanol 10 mg group (*p* = 0.00005) and 20 mg group (*p* = 0.0001) showed a significant difference compared to the placebo group. Within each policosanol group, multiple comparisons using the LSD method revealed the policosanol 10 mg group to show a significant reduction of LDL-C at 12 weeks (*p* < 0.001) compared to that at week 0. The policosanol 10 mg group also showed a significant decrease in LDL-C at 12 weeks (*p* = 0.001) compared to that at 4 weeks. The policosanol 20 mg group showed a more significant reduction of LDL-C at 4 weeks (*p* = 0.008), 8 weeks (*p* = 0.0002), and 12 weeks (*p* = 0.0001) compared to that at week 0. These results suggest that policosanol groups showed significant reduction of LDL-C with time dependent manner, while placebo group was not.

As shown in bottom panel of Table 5, from repeated measures ANOVA with %HDL-C data, the policosanol groups (10 mg, 20 mg) showed significant differences from the placebo in point of time (*p* < 0.001), and time × group interaction (*p* = 0.003). Further analysis with repeated measures ANOVA showed that both policosanol 10 mg (*p* < 0.001) and 20 mg (*p* < 0.001) groups showed significant differences within the group, whereas the placebo group showed no significance within the group (*p* = 0.311). One-way ANOVA between the groups showed no significant difference among the three groups at 0 week (*p*=.684), 4 weeks (*p* = 0.918), and 8 weeks (*p* = 0.769), except for 12 weeks (*p* = 0.035). Using the least significant difference (LSD) method as a post-hoc test with the 12-week data, the policosanol 10 mg group (*p* = 0.048) and 20 mg group (*p* = 0.014) showed a significant difference compared to the placebo group. Within each policosanol group, multiple comparisons using the LSD method revealed the policosanol 10 mg group to show a significant elevation of %HDL-C at 8 weeks (*p* = 0.002) and 12 weeks (*p* < 0.001) compared to that at week 0. The policosanol 10 mg group also showed a significant elevation in %HDL-C at 8 weeks (*p* = 0.026) and 12 weeks (*p* = 0.00007) compared to that at 4 weeks. The policosanol 20 mg group showed a more significant elevation of %HDL-C at 8 weeks (*p* = 0.00001), and 12 weeks (*p* = 0.00004) compared to that at week 0. The policosanol 20 mg group also showed a more significant elevation of %HDL-C at 8 weeks (*p* = 0.0002), and 12 weeks (*p* < 0.001) compared to that at week 4. These results strongly suggest that policosanol groups showed significant elevation of %HDL-C with time dependent manner, while placebo group was not. Serum aldosterone was remarkably decreased in the policosanol groups by 35% and 24% for the 10 and 20 mg groups, respectively, while the placebo group showed no change after 12-week consumption.

### 3.4. Correlation Study among Blood Pressure and Lipid Parameters

Pearson’s correlation analysis of peripheral and aortic blood pressure with lipid profile after 12 weeks of placebo and policosanol treatment is shown in the Appendix A. After 12 weeks of placebo group, peripheral blood pressure and aortic pressure were significantly correlated with MAP. However, in week 0, the peripheral DBP was negatively associated with % HDL-C. (Appendix A). After 12 weeks of therapy with 10 mg of policosanol, the correlation among peripheral, aortic pressure and the lipid profile was improved. The peripheral SBP was positively correlated with TG, TG/HDL, and MAP. Moreover, the peripheral DBP was negatively correlated with % HDL (Appendix A). In the same group, the aortic SBP was also positively correlated with TG/HDL and MAP (Appendix A). The correlation was improved for participants in the 20 mg policosanol group. A similar trend was seen for the correlation between different blood pressure measurements and lipid parameters in participants who consumed 20 mg of policosanol for 12 weeks, especially the peripheral blood pressure (Appendix A). The aortic blood pressure was correlated with MAP. In week 0, the peripheral DBP and aortic DBP were significant with TG, TG/HDL, and MAP (Appendix A).

## 4. Discussion

Hypertension is closely linked to the incidence of metabolic syndrome [23], which involves abdominal obesity, high serum TG level, low HDL-C level, and insulin resistance. Treatment of prehypertension is very important to reduce the risk of stroke, coronary artery disease, impairment of cognitive function, and chronic kidney disease [24]. It is also well known that hypertensive people have higher serum levels of TC and LDL-C and lower serum levels of HDL-C than normotensive subjects [7,25].

In the current study, at week 0, all participants in this study who completed 12 weeks consumption (*n* = 76) had similar BP (around 135.6 ± 9.1 mmHg of SBP and 84.6 ± 7.5 mmHg of DBP), serum TC (188 ± 20 mg/dL) and LDL-C (130 ± 32 mg/dL). However, policosanol consumption for 12 weeks lowered the peripheral BP, aortic BP, MAP, and serum levels of TC and LDL-C in a dose-dependent manner. In particular, the policosanol 20 mg group showed a remarkable reduction in peripheral SBP and DBP in a time-dependent manner between each group of different doses compared with the placebo group after 12 weeks from repeated measurement ANOVA (Table 3). The policosanol 20 mg group maintained the significantly lower DBP compared with placebo after 8 weeks. 

These outcomes are in good agreement with our recent reports, which showed that consumption of policosanol for 8 or 24 weeks lowered BP and serum TC and LDL-C via CETP inhibition and enhancement of HDL functionality [17,19]. The improvement in the serum lipid profile in the current study correlates well with enhancement of HDL functionality in our previous reports [17,19]. A contemporary meta-analysis of 22 studies also described that policosanol could be used to lower lipid levels and to safely elevate HDL-C levels [26]. The LDL-C lowering by policosanol can contribute to the decrease in aldosterone production via inhibition of scavenger receptor-mediated signaling as suggested previously [9,10]

After 8 weeks, the policosanol 10 mg group showed 3.4% reduction in peripheral DBP; however, the policosanol 20 mg group exhibited a 5.5% reduction in peripheral DBP (Figure 2). Interestingly, there was similar efficacy between the policosanol 10 mg and 20 mg groups to lower serum TC and LDL-C and raise HDL-C. Indeed, at 12 weeks, the policosanol 20 mg group (7.5% reduction from 0 week) showed greater reduction in MAP than the 10 mg group (5.2% reduction from 0 week). This result shows that policosanol improved not only peripheral BP, but also helped to alleviate aortic BP and MAP, especially 20 mg of policosanol. A higher MAP is correlated with higher incidence of hypertension [27,28] and metabolic syndrome [29].

## 5. Conclusions

In conclusion, 12-week consumption of policosanol resulted in significant reductions of peripheral SBP and DBP, aortic SBP and DBP, mean arterial pressure (MAP), and serum TC and LDL-C with elevation of %HDL-C. The BP-lowering efficacy of policosanol is also well correlated with reductions in serum levels of TG and TG/HDL-C.

## Figures and Tables

**Figure 1 ijerph-16-00809-f001:**
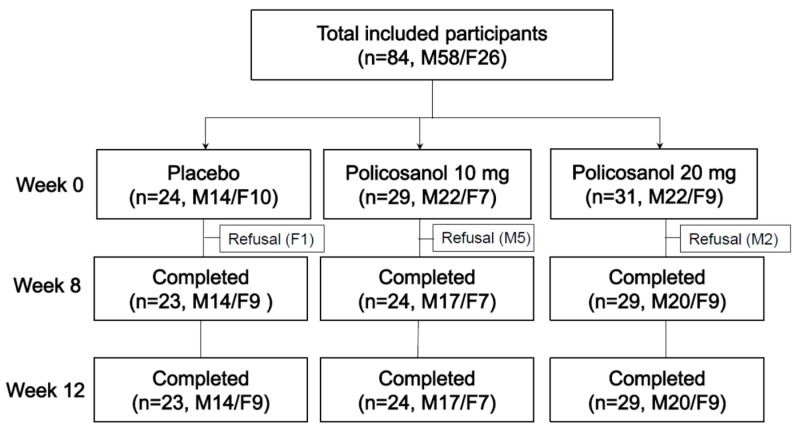
Design of study and participants. Inclusion criteria were normolipidemic, normoglycemic, and healthy subjects who had prehypertension (systolic 130–139 mmHg, diastolic 80–89 mmHg). M: Male; F: Female.

**Figure 2 ijerph-16-00809-f002:**
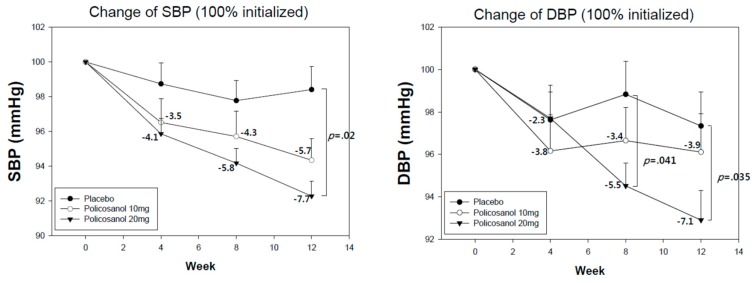
Change of peripheral blood pressures (SBP and DBP) during 12-week consumption of policosanol groups and placebo group from repeated measurement ANOVA. Data are expressed as mean ± SEM. SBP: systolic BP; DBP: diastolic BP; SEM: standard error of mean.

**Table 1 ijerph-16-00809-t001:** Change of blood pressures after 8-week policosanol consumption.

Variables		Group 1 Placebo (*n* = 23, M14/F9)		Group 2 Policosanol 10 mg (*n* = 24, M17/F7)		Group 3 Policosanol 20 mg (*n* = 29, M20/F9)	
Age		31.3 ± 14.2		32.4 ± 14.8		27.9 ± 10.2	
**Body composition**		**Week 0**	**Week 8**	**Week 0**	**Week 8**	**Week 0**	**Week 8**
BMI (body mass index)		22.0 ± 2.9	22.3 ± 2.8	23.0 ± 3.7	22.9 ± 3.7	23.5 ± 3.0	23.2 ± 2.8 *
Subcutaneous fat (kg)		11.9 ± 4.1	12.2 ± 4.2	13.5 ± 5.4	13.7 ± 5.7	13.6 ± 4.6	13.2 ± 4.3 *
Visceral fat (kg)		1.6 ± 1.1	1.7 ± 1.1	2.0 ± 1.1	2.1 ± 1.3	1.9 ± 1.0	1.8 ± 0.8 *
**Peripheral BP (mmHg)**		**Week 0**	**Week 8**	**Week 0**	**Week 8**	**Week 0**	**Week 8**
Average	Systolic	134.4 ± 8.8	131.3 ± 10.3	135.8 ± 12.3	129.8 ± 12.9 **	136.3 ± 6.1	128.4 ± 9.1 ***
	Diastolic	85.4 ± 6.8	84.3 ± 8.9	84.3 ± 8.5	81.3 ± 8.7 *	84.2 ± 7.3	79.5 ± 7.8 ***
**Aortic BP (mmHg)**		**Week 0**	**Week 8**	**Week 0**	**Week 8**	**Week 0**	**Week 8**
Aortic	Systolic	123.1 ± 11.8	117.1 ± 11.0 **	123.0 ± 12.5	114.6 ± 11.8 ***	120.7 ± 8.7	114.0 ± 11.2 ***
	Diastolic	90.2 ± 7.9	87.7 ± 8.5	88.5 ± 9.2	85.0 ± 8.8 *	88.6 ± 8.0	84.4 ± 9.5 **
Mean Arterial Pressure		101.7 ± 6.9	100.0 ± 9.1	101.5 ± 9.3	97.4 ± 9.8 *	101.6 ± 6.3	95.6 ± 7.6 ***

Data are expressed as mean ± SD. * *p* < 0.05 vs. week 0; ** *p* < 0.01 vs. week 0; *** *p* < 0.001 vs. week 0 in each group. M: male; F: Female; BP: blood pressure; SD: Standard deviation.

**Table 2 ijerph-16-00809-t002:** Change of blood pressures after 12-week policosanol consumption.

Variables		Group 1 Placebo (*n* = 23, M14/F9)		Group 2 Policosanol 10 mg (*n* = 24, M17/F7)		Group 3 Policosanol 20 mg (*n* = 29, M20/F9)	
Age		31.3 ± 14.2		32.4 ± 14.8		27.9 ± 10.2	
**Body composition**		**Week 0** *n* = 23	**Week 12** *n* = 23	**Week 0** *n* = 23	**Week 12** *n* = 23	**Week 0** *n* = 29	**Week 12** *n* = 29
BMI (body mass index)		22.0 ± 2.9	22.2 ± 2.9	23.0 ± 3.8	23.1 ± 3.9	23.5 ± 3.0	23.4 ± 2.8
Subcutaneous fat (kg)		11.9 ± 4.1	11.7 ± 3.9	13.5 ± 5.5	14.1 ± 5.6	13.6 ± 4.6	13.5 ± 4.3
Visceral fat (kg)		1.6 ± 1.1	1.6 ± 0.9	2.0 ± 1.2	2.1 ± 1.3	1.9 ± 1.0	1.9 ± 0.9
**Peripheral BP (mmHg)**		**Week 0** *n* = 23	**Week 12** *n* = 23	**Week 0** *n* = 24	**Week 12** *n* = 24	**Week 0** *n* = 29	**Week 12** *n* = 29
Average	Systolic	134.4 ± 8.8	132.1 ± 10.2	135.8 ± 12.3	127.7 ± 9.6 ***	136.3 ± 6.1	125.8 ± 8.7 ***
	Diastolic	85.4 ± 6.8	83.1 ± 9.1	84.3 ± 8.5	80.6 ± 7.3 *	84.2 ± 7.3	78.0 ± 7.7 ***
**Aortic BP (mmHg)**		**Week 0** *n* = 21	**Week 12** *n* = 21	**Week 0** *n* = 24	**Week 12** *n* = 24	**Week 0** *n* = 29	**Week 12** *n* = 29
Aortic	Systolic	122.1 ± 11.8	117.8 ± 9.0	123.0 ± 12.5	113.9 ± 8.6 ***	120.7 ± 8.7	110.7 ± 9.0 ***
	Diastolic	90.2 ± 8.3	87.0 ± 9.3	88.5 ± 9.2	85.5 ± 7.5	88.6 ± 8.0	84.1 ± 9.2 **
Mean Arterial Pressure		101.7 ± 6.9	99.4 ± 9.1	101.5 ± 9.3	96.3 ± 7.6 **	101.6 ± 6.3	94.0 ± 7.6 ***

Data are expressed as mean ± SD. * *p* < 0.05 vs. week 0; ** *p* < 0.01 vs. week 0; *** *p* < 0.001 vs. week 0 in each group.

**Table 3 ijerph-16-00809-t003:** Repeated measures ANOVA of peripheral systolic BP and diastolic BP between the three groups.

Variables	Groups	0 Week ^a^	4 Week ^b^	8 Week ^c^	12 Week ^d^	*p* for Time Differences	F (*p*)	Sources	F	*p*
		Mean ± SD	Mean ± SD	Mean ± SD	Mean ± SD					
Systolic BP	Placebo (*n* = 23)	134.4 ± 8.8 ^a^	132.7 ± 10.7 ^b^	131.3 ± 10.3 ^c^	132.1 ± 10.2 ^d^	No significance	1.292 (0.286)	Group	0.438	0.647
	Policosanol 10 mg (*n* = 24)	135.8 ± 12.3 ^a^	130.8 ± 12.0 ^b^	129.8 ± 12.9 ^c^	127.7 ± 9.6 ^d^	a > b (0.0004) a > c (0.007) a > d (0.0002) b > d (0.025)	7.675 (<0.001)	Time	25.44	<0.001
	Policosanol 20 mg (*n* = 29)	136.3 ± 6.1 ^a^	130.7 ± 9.0 ^b^	128.2 ± 9.0 ^c^	125.9 ± 8.6 ^d^	a > b (<0.001) a > c (<0.001) a > d (<0.001) b > c (0.021) b > d (<0.001) c > d (0.007)	35.687 (<0.001)	Time × Group	2.81	0.012
	*p* for group differences	0.753	0.770	0.631	0.020 (Placebo vs. Policosanol 20 mg)					
Diastolic BP	Placebo (*n* = 23)	85.4 ± 6.8 ^a^	83.3 ± 8.9 ^b^	84.3 ± 8.9 ^c^	83.1 ± 9.1 ^d^	No significance	1.408 (0.270)	Group	1.277	0.285
	Policosanol 10 mg (*n* = 24)	84.3 ± 8.5 ^a^	80.8 ± 7.5 ^b^	81.3 ± 8.7 ^c^	80.6 ± 7.3 ^d^	a > b (0.021) a > c (0.042) a > d (0.028)	3.381 (0.028)	Time	8.408	<0.001
	Policosanol 20 mg (*n* = 29)	84.2 ± 7.3 ^a^	82.0 ± 6.9 ^b^	79.2 ± 7.6 ^c^	78.0 ± 7.7 ^d^	a > b (0.002) a > c (<0.001) a > d (<0.001) b > c (0.002) b > d (<0.001)	11.273 (<0.001)	Time × Group	3.377	0.023
	*p* for group differences	0.831	0.532	0.041 (Placebo vs. Policosanol 20 mg)	0.035 (Placebo vs. Policosanol 20 mg)					

^a^ data of 0 week; ^b^ data of 4 week; ^c^ data of 8 week; ^d^ data of 12 week.

**Table 4 ijerph-16-00809-t004:** Change of serum lipid profile after 12-week policosanol consumption.

Variables	Group 1 Placebo (*n* = 23, M14/F9)		Group 2 Policosanol 10 mg (*n* = 23, M16/F7)		Group 3 Policosanol 20 mg (*n* = 28, M19/F9)	
Blood Profile	Week 0	Week 12	Week 0	Week 12	Week 0	Week 12
TC (mg/dL)	188.1 ± 35.1	203.7 ± 40.3	183.6 ± 29.9	166.1 ± 30.0 *	186.3 ± 42.5	170.3 ± 31.8 *
TG (mg/dL)	87.4 ± 35.2	80.5 ± 38.9	104.6 ± 70.6	116.5 ± 81.1	108.3 ± 92.6	105.5 ± 73.8
HDL-C (mg/dL)	39.8 ± 6.9	43.5 ± 7.4	35.9 ± 6.3	41.7 ± 7.9 **	36.9 ± 9.4	44.2 ± 7.6 **
% HDL-C in TC	21.9 ± 5.8	22.2 ± 6.0	20.2 ± 4.7	25.9 ± 6.5 **	21.3 ± 8.4	26.6 ± 6.0 **
TG/HDL-C	2.3 ± 1.2	2.0 ± 1.2	3.2 ± 2.7	3.0 ± 2.5	3.4 ± 3.6	2.4 ± 1.6
LDL-C (mg/dL)	130.8 ± 35.7	144.1 ± 37.6	126.7 ± 27.5	101.2 ± 31.3 **	127.8 ± 39.1	105.0 ± 32.1 **
LDL-C/HDL-C	3.4 ± 1.2	3.4 ± 1.1	3.6 ± 1.0	2.5 ± 1.1 **	3.8 ± 1.6	2.5 ± 0.9 **
Glucose (mg/dL)	86.3 ± 12.6	91.3 ± 11.1	97.3 ± 11.0	96.3 ± 12.6	93.5 ± 14.0	95.7 ± 14.5
Aldosterone (ng/dL)	19.7 ± 13.9	20.0 ± 12.6	30.0 ± 23.1	19.3 ± 9.3 *	24.7 ± 12.1	19.0 ± 9.1 *

Data are expressed as mean ± SD, * *p* < 0.05 vs. week 0; ** *p* < 0.001 vs. week 0 in each group. TC, total cholesterol; TG, triglyceride; HDL-C, High-density lipoprotein-cholesterol; LDL-C, low-density lipoprotein-cholesterol.

**Table 5 ijerph-16-00809-t005:** Repeated measures ANOVA of serum TC, LDL-C, and %HDL-C profile between the three groups.

Variables	Groups	0 Week ^a^	4 Week ^b^	8 Week ^c^	12 Week ^d^	*p* for Time Differences	F (*p*)	Sources	F	*p*
		Mean ± SD	Mean ± SD	Mean ± SD	Mean ± SD					
TC	Placebo (*n* = 23)	188.1 ± 35.1 ^a^	188.0 ± 30.3 ^b^	188.9 ± 39.2 ^c^	203.7 ± 40.3 ^d^	No significant	1.683 (0.203)	Group	2.980	0.057
	Policosanol 10 mg (*n* = 23)	183.6 ± 29.9 ^a^	178.6 ± 33.6 ^b^	174.9 ± 29.1 ^c^	166.1 ± 30.0 ^d^	a > d (0.001) b > d (0.010)	4.994 (0.010)	Time	2.389	0.076
	Policosanol 20 mg (*n* = 28)	186.3 ± 42.5 ^a^	171.4 ± 29.1 ^b^	170.2 ± 37.2 ^c^	170.3 ± 31.8 ^d^	a > b (0.005) a > c (0.015) a > d (0.029)	3.737 (0.024)	Time × Group	2.997	0.009
	*p* for group differences	0.915	0.169	0.167	0.0005 (Placebo vs. Policosanol 10 mg, Policosanol 20 mg)					
LDL-C	Placebo (*n* = 23)	130.8 ± 35.7 ^a^	126.8 ± 30.5 ^b^	125.0 ± 26.6 ^c^	144.1 ± 37.6 ^d^	No significant	2.346 (0.103)	Group	3.507	0.035
	Policosanol 10 mg (*n* = 23)	126.7 ± 27.5 ^a^	120.4 ± 30.9 ^b^	111.0 ± 29.1 ^c^	101.2 ± 31.3 ^d^	a > d (<0.001) b > d (0.001)	8.509 (0.001)	Time	7.472	<0.001
	Policosanol 20 mg (*n* = 28)	127.8 ± 39.1 ^a^	112.9 ± 24.9 ^b^	107.1 ± 40.1 ^c^	105.0 ± 32.1 ^d^	a > b (0.008) a > c (0.0002) a > d (0.0001)	7.406 (<0.001)	Time × Group	4.537	<0.001
	*p* for group differences	0.916	0.226	0.148	<0.001 (Placebo vs. Policosanol 10 mg, Policosanol 20 mg)					
%HDL-C in TC	Placebo (*n* = 23)	21.9 ± 5.8 ^a^	21.4 ± 5.1 ^b^	24.1 ± 7.5 ^c^	22.2 ± 6.0 ^d^	No significant	1.271 (0.311)	Group	0.289	0.750
	Policosanol 10 mg (*n* = 23)	20.2 ± 4.7 ^a^	21.7 ± 7.2 ^b^	25.2 ± 8.2 ^c^	25.9 ± 6.5 ^d^	a < c (0.002) a < d (<0.001) b < c (0.026) b < d (0.00007)	18.749 (<0.001)	Time	25.886	<0.001
	Policosanol 20 mg (*n* = 28)	21.3 ± 8.4 ^a^	21.0 ± 4.9 ^b^	25.7 ± 8.3 ^c^	26.6 ± 6.0 ^d^	a < c (0.00001) a < d (0.00004) b < c (0.0002) b < d (<0.001)	16.887 (<0.001)	Time × Group	3.453	0.003
	*p* for group differences	0.684	0.918	0.769	0.035 (Placebo vs. Policosanol 10 mg, Policosanol 20 mg)					

Data are expressed as mean ± SD. TC, total cholesterol; HDL-C, High-density lipoprotein-cholesterol; LDL-C, low-density lipoprotein-cholesterol. ^a^ data of 0 week; ^b^ data of 4 week; ^c^ data of 8 week; ^d^ data of 12 week.

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
