# Peer review of "Short-Term Consumption of Cuban Policosanol Lowers Aortic and Peripheral Blood Pressure and Ameliorates Serum Lipid Parameters in Healthy Korean Participants: Randomized, Double-Blinded, and Placebo-Controlled Study"

_ijerph, 2019, doi:10.3390/ijerph16050809_

Round 1
Reviewer 1 Report
The authors addressed all of the issues raised in the original review. There is no other concern.
Reviewer 2 Report
My concerns have been satisfactorily addressed.
This manuscript is a resubmission of an earlier submission. The following is a list of the peer review reports and author responses from that submission.
Round 1
Reviewer 1 Report
In this manuscript Park et al report a randomized, double-blind study on Korean participants with pre-hypertension to test the positive effects of policosanol on blood pressure and lipid profile, within 12 weeks of treatment. The authors show a clear positive effect of policosanol, both after 8 and 12 weeks of consumption, on blood pressure as well as plasma lipid composition.
I found the manuscript very interesting, with a clear study design and results supported by the data generated.
I only have minor points:
· Pre-hypertension is either written with the hyphen or as a single word within the manuscript. I recommend to use one or the other.
· For sake of clarity, I would describe the composition of the placebo already in the 2.1 section.
· In the Introduction paragraph, line 46, authors write high density lipoprotein and then HDL-C in brackets. I guess they meant HDL-cholesterol. Same in line 5 of the discussion section.
· Authors show an 8.2% increase in TC at week 12 for the placebo group. Is that significant?
· Participants are both male and female subjects. Since there is often a gender difference in the metabolism of lipoproteins, I wonder if there is a difference in the response to policosanol between male and female subjects regarding plasma lipid composition.
Author Response
Response to Review Report 1:
Thank you for your valuable review and comments.
In this manuscript Park et al report a randomized, double-blinded study on Korean participants with pre-hypertension to test the positive effects of policosanol on blood pressure and lipid profile, within 12 weeks of treatment. The authors show a clear positive effect of policosanol, both after 8 and 12 weeks of consumption, on blood pressure as well as plasma lipid composition.
I found the manuscript very interesting, with a clear study design and results supported by the data generated.
Thank you so much for your positive feedback.
I only have minor points:
· Pre-hypertension is either written with the hyphen or as a single word within the manuscript. I recommend to use one or the other.
We would use only prehypertension. We have modified the word as per the reviewer suggestion in the revised version of the manuscript. Thank you.
· For sake of clarity, I would describe the composition of the placebo already in the 2.1 section.
We have inserted a composition of the placebo in the section 2.1 as per reviewer’s suggestion as below Table.
Placebo( 200mg/tablet) | the weight of ingredient |
Policosanol | 0 |
lactose | 107.6mg |
Cellulose, Microcrystalline | 85.0mg |
Magnesium stearate | 1.6mg |
Hydroxypropyl methyl cellulose | 4.0mg |
Glycerin fatty acid ester | 0.4mg |
Titanium dioxide | 0.28mg |
Gardenia Blue color | 1.0mg |
“The placebo tablet had the same taste and smell with the identical color. It contained the same basic pigment (Gardenia Blue color) and ingredients such as lactose, cellulose, glycerine fatty acid ester, magnesium stearate, etc, except policosanol”.
· In the Introduction paragraph, line 46, authors write high density lipoprotein and then HDL-C in brackets. I guess they meant HDL-cholesterol. Same in line 5 of the discussion section.
We are really thankful to the reviewer for the comment. Yes, the meaning was same here to denote HDL-C, we have revised this in the whole manuscript.
· Authors show an 8.2% increase in TC at week 12 for the placebo group. Is that significant?
In the placebo group the increase of TC was not found to be significant at the week 12. Thank you
· Participants are both male and female subjects. Since there is often a gender difference in the metabolism of lipoproteins, I wonder if there is a difference in the response to policosanol between male and female subjects regarding plasma lipid composition.
Thank you so much for the thought-provoking comments. We fully agree with the reviewer comments that there might be a gender difference in lipoprotein metabolism. However, our recruited subjects were healthy with blood pressure in the prehypertensive stage. We did not find any significant difference based on gender that affected the plasma lipid composition as in our previous report with women [1].
[1]. Cho, K.H.; Kim, S.J.; Yadav, D.; Kim, J.Y.; Kim, J.R. Consumption of Cuban Policosanol Improves Blood Pressure and Lipid Profile via Enhancement of HDL Functionality in Healthy Women Subjects: Randomized, Double-Blinded, and Placebo-Controlled Study.. Oxid Med Cell Longev. 2018, 2018:4809525.
Reviewer 2 Report
In this manuscript, Hye-Jeong Par and co-authors examined the effect of Cuban policosanol on central and peripheral blood pressure as well as serum lipid profile in healthy Korean participants. In general, it is a study with novelty, since compared to the previous study by the same group (Kim, S.J et.al Long-Term Consumption of Cuban Policosanol Lowers Central and Brachial Blood Pressure and Improves Lipid Profile With Enhancement of Lipoprotein Properties in Healthy Korean Participants Front Physiol. 2018, 9, 412), this study added the result from short-term of policosanol treatment. However, the conclusion is not convincing because of the improper statistical analyses. Several major concerns are listed as follows:
Is this an original clinical trial or a substudy analysis of existing data?
I found that a published paper seems from the study.
Is the sample size large enough to detect a clinically significant difference? Please add a paragraph about sample size and power estimation.
The author compared the change of BP from baseline to week 8 and baseline to week 12 separately for all three groups. The author cannot claim that Policosanol is effective based on the evidence that significant decrease observed in treatment group but not in Placebo group. An appropriate analysis should be assessing the difference between placebo group and treatment group. Moreover, BP measurements at different time points from one patient are correlated. The author should use a repeated measurement analysis.
Did the authors use a paired Student t test in Table 1, 2 and 3 to compare all variables between baseline and after policosanol treatment? The author should include more information in the part of data analysis.
Figure 2 did compare the difference between placebo and Policosanol 20mg group. As major evidence to support the conclusion, Figure 2 should be presented in the result part but not in discussion part. The authors should also compare 10mg group to placebo group. Please also clarify the statistical model in method part.
The authors conclude “The BP-lowering efficacy of policosanol is also associated with reductions of serum TC, LDL-C, and aldosterone as well as elevation of HDL-C.” However, the authors didn’t perform any correlations analysis between BPs and serum lipid profiles for baseline or after policosanol treatment.
Author Response
Response to Review Report 2:
In this manuscript, Hye-Jeong Park and co-authors examined the effect of Cuban policosanol on central and peripheral blood pressure as well as serum lipid profile in healthy Korean participants. In general, it is a study with novelty, since compared to the previous study by the same group (Kim, S.J et.al Long-Term Consumption of Cuban Policosanol Lowers Central and Brachial Blood Pressure and Improves Lipid Profile With Enhancement of Lipoprotein Properties in Healthy Korean Participants Front Physiol. 2018, 9, 412), this study added the result from short-term of policosanol treatment. However, the conclusion is not convincing because of the improper statistical analyses. Several major concerns are listed as follows:
Is this an original clinical trial or a substudy analysis of existing data?
Answer) This study was mainly focused on the blood pressure lowering effect of short-term consumption of policosanol in Korean participants with prehypertension. Therefore, we used a sub-study analysis of existing data. However, we have included more participants to validate the blood pressure lowering effect in human subjects.
I found that a published paper seems from the study.
Answer) Thank you so much for pointing out our previous paper that was published in Front Physiol. 2018. However, we used that data to monitor the long-term effect of policosanol in prehypertensive participants in terms of the glycation and oxidation of lipoproteins. Due to the long-term (24 weeks) consumption, there were fewer participants who completed that study than those that completed the 12-week study. Therefore, we need to compare the blood pressure lowering effect at 8 weeks and 12 weeks with more participants to get more accurate and reliable statistical significance. Also, the previous paper focused on both the BP lowering effect and the enhancement of HDL functionality with lipid-lowering effects. The previous paper suggested that policosanol improved the antioxidant and anti-glycation activity along with the long-term blood pressure lowering and lipid improvement.
The focus of this paper was to analyze the short-term effect of policosanol on blood pressure improvement in healthy participants. We observed a reduction in lipid profile especially in LDL level. A sufficient number of participants were used in the study analysis. The uniqueness of this study is that we have compared the changes of peripheral blood pressures (SBP and DBP) during 12-week consumption of policosanol (20 mg) or placebo.
Is the sample size large enough to detect a clinically significant difference? Please add a paragraph about sample size and power estimation.
Answer) The authors are thankful to the kind reviewer for the comments on the clinically significant difference based on the study. The sample size is sufficient to detect a significant difference in blood pressure and lipid parameters after the consumption of policosanol by the recruited participants.
Sample size was calculated by the following formula.
1. Sample size calculation based on serum lipid:
Initially we focused on both lipid and blood pressure, as these risk factors are associated with a large number of comorbidities in various disease conditions. We have calculated the power estimation based on our previous study [1]. Sample size was estimated to detect the difference between two means using the formula below, for instances considering the mean+SD of LDL Cholesterol for the placebo and Policosanol groups as 137±20 mg/dl and 81±7 mg/dl, respectively [1], with alpha error of 5%, power of 20% and absolute error of margin of 9 mg/dl. The minimum sample size per group was calculated and found to be 21. Assuming 10% attrition rate, the final minimum sample size per group was decided as 23.
Z1-α/2 is table value from the standard normal distribution corresponding to area
D represents the absolute precision required on either side of the true value
σ is the average of the standard deviations in the study groups for a particular variable.
Retrospective power estimation:
Power was estimated using online software [Reference: http://powerandsamplesize.com/] to detect the difference between two means, considering the mean+SD of LDL Cholesterol after 12 weeks of treatment for the placebo and Policosanol (10mg) groups as 144.1±37.6 mg/dl and 101.2±31.3 mg/dl, respectively. The estimated power was 95.48%.
Power was estimated using online software [Reference: http://powerandsamplesize.com/] to detect the difference between two means, considering the mean+SD of LDL Cholesterol after 12 weeks of treatment for the placebo and Policosanol (20mg) groups as 144.1±37.6 mg/dl and 105.0±32.1 mg/dl, respectively. The estimated power was 90.07%.
[1]. Kim, S.J.; Yadav, D.; Kim, J.R. Cho, K.H. Long-term consumption of Cuban policosanol lowers central and brachial blood pressure and improves lipid profile with enhancement of lipoprotein properties in healthy Korean participants (Frontiers in Physiology April 2018 | Volume 9 | Article 412
The author compared the change of BP from baseline to week 8 and baseline to week 12 separately for all three groups. The author cannot claim that Policosanol is effective based on the evidence that significant decrease observed in treatment group but not in Placebo group. An appropriate analysis should be assessing the difference between placebo group and treatment group. Moreover, BP measurements at different time points from one patient are correlated. The author should use a repeated measurement analysis.
Answer) The authors are thankful to receive such thought-provoking comments, and the authors agree with the reviewer comments. The main reason for this was to make our study design placebo controlled. We observed a significant correlation between the policosanol group (treatment) and lipid parameters. In figure 2 we have reported the effects of weekly consumption of policosanol and placebo on SBP and DBP. We accept that BP measurements at different time points from one patient are correlated and we need a repeated measurement analysis. The result of repeated measurement analysis has been documented below. Additionally, the measurement of blood pressure in the study used three different devices and we used average values to calculate the peripheral blood pressure.
As per reviewer’s suggestion, we did repeated measurement analysis as below.
<Repeated measurement ANOVA analysis>
1. Peripheral SBP
Within-Subjects Factors | |
Measure: | |
every4weeks | Dependent Variable |
1 | Peripheral SBP 0 week |
2 | Peripheral SBP After_4 week |
3 | Peripheral SBP After_8 week |
4 | Peripheral SBP After_12 week |
Since Mauchly’s test of sphericity was satisfied to be significant (p=0.082), therefore, tests of Within-Subjects Effects were analyzed as below Table.
Between-Subjects Factors | |||
Value Label | N | ||
group | 1.00 | Placebo | 23 |
2.00 | Policosanol 10 mg | 24 | |
3.00 | Policosanol 20 mg | 29 | |
Tests of Within-Subjects Effects | ||||||
Measure: | ||||||
Source | Type III Sum of Squares | df | Mean Square | F | Sig. | |
Every 4 weeks | Sphericity Assumed | 2067.678 | 3 | 689.226 | 25.443 | .000 |
Greenhouse-Geisser | 2067.678 | 2.757 | 749.964 | 25.443 | .000 | |
Huynh-Feldt | 2067.678 | 2.955 | 699.834 | 25.443 | .000 | |
Lower-bound | 2067.678 | 1.000 | 2067.678 | 25.443 | .000 | |
Every 4 weeks * group | Sphericity Assumed | 456.463 | 6 | 76.077 | 2.808 | .012 |
Greenhouse-Geisser | 456.463 | 5.514 | 82.781 | 2.808 | .015 | |
Huynh-Feldt | 456.463 | 5.909 | 77.248 | 2.808 | .012 | |
Lower-bound | 456.463 | 2.000 | 228.231 | 2.808 | .067 | |
Error (every 4 weeks) | Sphericity Assumed | 5932.435 | 219 | 27.089 | ||
Greenhouse-Geisser | 5932.435 | 201.264 | 29.476 | |||
Huynh-Feldt | 5932.435 | 215.681 | 27.506 | |||
Lower-bound | 5932.435 | 73.000 | 81.266 | |||
- Since p<0.01 for every 4 weeks, all peripheral SBP were significantly different within each group and decreased in a time-dependent manner.
-Since p<0.05 for every 4 weeks*group, all peripheral SBP were significantly different in a time-dependent manner between each group of different doses.
2. Peripheral DBP
Within-Subjects Factors | |
Measure: | |
Every 4 weeks | Dependent Variable |
1 | Peripheral DBP 0 week |
2 | Peripheral DBP After_4 week |
3 | Peripheral DBP After_8 week |
4 | Peripheral DBP After_12 week |
Between-Subjects Factors | |||
Value Label | N | ||
Group | 1.00 | Placebo | 23 |
2.00 | Policosanol 10 mg | 24 | |
3.00 | Policosanol 20 mg | 29 | |
Since Mauchly’s test of sphericity was not satisfied to be significant (p=0.011), therefore, tests of Multivariate Testsa was analyzed as below Table.
Multivariate Testsa | ||||||
Effect | Value | F | Hypothesis df | Error df | Sig. | |
Every 4 weeks | Pillai's Trace | .262 | 8.408b | 3.000 | 71.000 | .000 |
Wilks' Lambda | .738 | 8.408b | 3.000 | 71.000 | .000 | |
Hotelling's Trace | .355 | 8.408b | 3.000 | 71.000 | .000 | |
Roy's Largest Root | .355 | 8.408b | 3.000 | 71.000 | .000 | |
Every 4 weeks * Group | Pillai's Trace | .139 | 1.792 | 6.000 | 144.000 | .105 |
Wilks' Lambda | .863 | 1.810b | 6.000 | 142.000 | .101 | |
Hotelling's Trace | .157 | 1.826 | 6.000 | 140.000 | .098 | |
Roy's Largest Root | .141 | 3.377c | 3.000 | 72.000 | .023 | |
a. Design: Intercept + Group | ||||||
b. Exact statistic | ||||||
c. The statistic is an upper bound on F that yields a lower bound on the significance level. | ||||||
- Since p<0.01 for every 4 weeks, all peripheral DBP were significantly different within each group and decreased in a time-dependent manner.
-Since p<0.05 from Roy's Largest Root for every 4 weeks*group, all peripheral DBP were significantly different in a time-dependent manner between each group of different doses.
Did the authors use a paired Student t test in Table 1, 2 and 3 to compare all variables between baseline and after policosanol treatment? The author should include more information in the part of data analysis.
Answer) Thank you for your comments. We used paired student t-test while analyzing our study data. As per your suggestion, we have updated the data analysis in the result section as below:
“We also quantify the confidence interval (CI, usually 95%) of blood pressure values in the different group to measure a range that provides the variability of observed population statistic (precision) and also to report the probable relationship in the population from where the data was extracted (accuracy). For the placebo group, the measured CI at week 0 and week 8 for the peripheral and aortic pressure were [SBP 95% (-.24, 6.42), (1.68, 10.2) DBP 95% (-1.78, 3.87), (-.08, 5.12). For the policosanol, 10 mg group, the measured CI at week 0 and week 8 for the peripheral and aortic pressure were [SBP 95% (1.81, 10.2), (4.26, 12.4) DBP 95% 0.12, 5.96), (0.40, 6.76). For the policosanol, 20 mg group, the measured CI at week 0 and week 8 for the peripheral and aortic pressure were [SBP 95% (5.53, 10.26), (3.54, 9.76) DBP 95% (2.85, 6.52), (1.28, 7.06)”.
“For the placebo group, the measured CI at week 0 and week 12 for the peripheral and aortic pressure were [SBP 95% (-1.52, 6.13), (-.077, 8.83) DBP 95% (-.65, 5.28), (-.38, 6.86). For the policosanol, 10 mg group, the measured CI at week 0 and week 12 for the peripheral and aortic pressure were [SBP 95% (4.25, 11.91), (5.46, 12.61) DBP 95% (0.44, 6.96), (-.817, 6.81). For the policosanol, 20 mg group, the measured CI at week 0 and week 12 for the peripheral and aortic pressure were [SBP 95% (8.06, 12.96), (7.31, 12.6) DBP 95% (3.72, 8.55), (1.20, 7.83)”.
Figure 2 did compare the difference between placebo and Policosanol 20mg group. As major evidence to support the conclusion, Figure 2 should be presented in the result part but not in discussion part. The authors should also compare 10mg group to placebo group. Please also clarify the statistical model in method part.
Answer) We are grateful for the suggestion by the reviewer related to comparing the difference between placebo and Policosanol 20mg group. We have included a paragraph related to Fig. 2 in the result section. As far as the concern related to comparing the 10 mg group to placebo, we performed that comparison also but the result was not significant. Independent two sample t-test was used to compare the placebo and policosanol 20 mg group. The following paragraph was inserted into the Results section per the reviewer’s suggestion.
“Fig. 2 depicts the changes in the peripheral blood pressures (SBP and DBP) during 12-week consumption of policosanol 20 mg and placebo groups. Participants consumed with policosanol 20 mg resulted in a prominent reduction in peripheral SBP and DBP in a time-dependent manner. The Fig. 2 compared the placebo group with, the policosanol 20 mg group in weekly and demonstrated a straightforward reduction of SBP and DBP after 12 weeks. For comparing the placebo and policosanol group (20 mg) the study used the Independent two sample t-test”.
The authors conclude “The BP-lowering efficacy of policosanol is also associated with reductions of serum TC, LDL-C, and aldosterone as well as elevation of HDL-C.” However, the authors didn’t perform any correlations analysis between BPs and serum lipid profiles for baseline or after policosanol treatment.
Answer) We provided the correlation analysis between the parameters undertaken for consideration in the statistical measurement in the supplementary files. We also added a paragraph related to the query in the result section of the revised version of the manuscript.
“Correlation study among blood pressure and lipid parameters”
Pearson’s correlation analysis of peripheral and aortic blood pressure with lipid profile after 12 weeks of placebo and policosanol treatment shown in supplementary Tables (Table S1, Table S2, Table S3). After 12 weeks of placebo therapy, peripheral blood pressure and aortic pressure were significantly correlated with MAP. However, at the 0 week, the peripheral DBP was negatively correlated with % HDL-C. (Table S1). After 12 weeks of therapy with 10mg of policosanol, the correlation among peripheral, aortic pressure and the lipid profile was found to be improved. The peripheral SBP was positively correlated with TG, TG/HDL, and MAP. Moreover, the peripheral DBP was negatively correlated with % HDL (Table S2). In the same group, the Aortic SBP also demonstrated a positive correlation with TG/HDL and MAP (Table S2). The correlation study was improved for the participant belonging to 20 mg policosanol group. A Similar trend was seen when the correlation observed between different blood pressure and lipid parameters in the participants consumed with 20 mg of policosanol after 12 weeks of therapy, especially the peripheral blood pressure. (Table S3). The aortic blood pressure was correlated with MAP. At the week 12, the peripheral SBP was found to be significant with serum TG, TG/HDL, and MAP (Table S3)”.
As per reviewer’s comment, we also corrected the sentence in Conclusion as below.
“The BP-lowering efficacy of policosanol is also associated with reductions in serum TG and TG/HDL-C”.
Once again thank you so much for providing critical comments on our manuscript.
Round 2
Reviewer 2 Report
The manuscript has been significantly improved. Most of the issues raised in the original review have been well addressed. The authors correctly chose repeated measure ANOVA to assess the effect. However, the test is not done properly. Instead of reporting time effect, the author should also estimate the treatment effect and treatment*time interaction. If the interaction is significant, the authors need to determine the difference at each time point. If the interaction is not significant, the authors should report the overall main effect of treatment. The link below is a tutorial about how to perform repeated measure ANOVA in SPSS. (https://statistics.laerd.com/spss-tutorials/two-way-repeated-measures-anova-using-spss-statistics.php) I hope the authors could at least try to analyze the data in the most suitable method and show what the result is.